# A treatment strategy for meeting life as it is. Patients' and therapists' experiences of brief therapy in a district psychiatric centre: A qualitative study

Hilde V. Markussen [1,2]*, Lene Aasdahl[3,4], Petter Viksveen[5], Berith Hedberg[6], Marit B. Rise[1,2]

1 Department of Mental Health, Faculty of Medicine and Health Sciences, Norwegian University of Science and Technology, Trondheim, Norway, 2 St. Olavs Hospital, Trondheim University Hospital, Nidaros District Psychiatric Centre, Trondheim, Norway, 3 Department of Public Health and Nursing, Faculty of Medicine and Health Sciences, Norwegian University of Science and Technology, Trondheim, Norway, 4 Unicare Helsefort Rehabilitation Centre, Rissa, Norway, 5 SHARE–Centre for Resilience in Healthcare, Department of Quality and Health Technology, Faculty of Health Sciences, University of Stavanger, Stavanger, Norway, 6 IMPROVE Research Group, School of Health and Welfare, Jönköping University, Jönköping, Sweden

* hilde.v.markussen@ntnu.no

**Data Availability Statement:** There are both ethical and legal restrictions on sharing the data set due to sensitive information. It is not possible to publish

## Abstract

### Background

Young adults increasingly seek help for mental health problems. In 2016, a district psychiatric centre in Norway started a brief treatment program to provide early and effective help for moderate depression and anxiety.

### Aim

Exploring patients' and therapists' experiences of brief therapy, especially how the time limitation influences the treatment process.

### Methods

Individual interviews with 12 patients and focus group interviews with eight therapists analyzed using systematic text condensation.

### Results

The results constitute five themes: (1) Time-limit as a frame for targeted change, (2) Clarifying expectations and accountability, (3) Shared agreement on a defined treatment-project, (4) Providing tools instead of searching for causes, and (5) Learning to cope—not being cured.

### Conclusion

Time-limitation in brief therapy appeared to play a positive role, helping the therapists to structure the therapeutic process and strengthening patients' motivation. Shared

the original data as participants were guaranteed that their interviews would not be made publicly available. Therefore, data publication would violate their privacy rights and conflict with the General Data Protection Regulation (GDPR). The approved information letter to participants in the present study stated; The researchers have a duty of confidentiality, and information from the interviews will not be given to employees at the DPC nor anyone else. This study was approved by the "Regional Committee for Medical and Health Research Ethics" (REK) in central Norway (2018/49). REK approved the information letter before we used it in the study. Email: rek-midt@mh.ntnu.no The committee secretariats can also be contacted by telephone or office address which can be found at the online portal: https://rekportalen.no.

**Funding:** The first author (HVM) and the last author (MBR) received the funding through their affiliation with NTNU and St.Olavs Hospital. The Liaison Committee (No; Samarbeidsorganet) between the Central Norway Regional Health Authority and Norwegian University of Science and Technology (NTNU) funded the work under Grant no. 22314. The funder website: https://helse-midt.no/samarbeidsorganet The funders had no role in study design, data collection and analysis, decision to publish, or preparation of the manuscript.

**Competing interests:** The authors have declared that no competing interests exist.

**Abbreviations:** DPC, District Psychiatric Centre; CBT, Cognitive Behavioral Treatment.

understanding and activation during brief therapy may reinforce patients' responsibility and expectations to achieve individual goals. Brief therapy can be viewed as the start of a personal process towards "mastering life as it is". More research is needed to investigate the patients' long-term outcomes after treatment and to shed light on the potential for, and limitations of, mastering everyday-life.

## Introduction

The demand for mental health services is growing worldwide [1]. In addition to the suffering of patients and their families, mental illness causes major challenges in health care and society in terms of health and social care costs [2]. Despite substantial research advances showing what can be done to prevent and treat mental disorders and to promote mental health, the global burden of disease attributable to mental disorders has risen in all countries the last decades [3]. Different innovative and effective preventive and therapeutic strategies have been considered to reduce the burden of mental illness [4]. Still, investments in mental health care have shown to be insufficient in all countries, and disproportionate to the burden of mental health conditions the last decades [3]. Thus, there may be a need for further research on and financing of prevention and adequate treatment of various mental disorders in mental health services.

As early as the 1980s, time-limited therapies were presented as a possible solution for mental health services to deal with an increasing number of patients seeking help [5]. Based on the relationship between dose and treatment effect, a time-limit in outpatient treatment was both recommended and introduced into clinical practice in several countries in the following years [6]. Studies investigating the relationship between treatment length and improvement rate have shown that small treatment doses are associated with rapid change, and that improvement often takes place very early in the treatment course [7]. However, some studies have shown that there are large differences between patients with regards to their response to treatment, indicating that a fixed duration of treatment is inappropriate [8] and that a tailoring of treatment length to individual patients is necessary [9]. This means that a larger number of treatment sessions does not necessarily give better results than shorter treatment programs [8, 10]. It has been argued that treatment should end when the patient has improved to a "sufficient" degree, which is determined by the patient's ability to manage life outside therapy [7, 8].

Brief therapy is defined as an innovative and efficient mental health therapy approach, focusing on the present ("here and now") and the patient's strengths [11]. The words short-term, time-limited, and brief therapy are used interchangeably in the literature [11]. Limitation of therapy processes was initially developed as a contrast to long and comprehensive psychoanalytic treatment [11]. Short-term psychodynamic psychotherapy has shown promising benefits for adults with common mental disorders [12], and short-term cognitive behavioral treatment (CBT) may be effective in treating depression and anxiety [13]. While the real-world clinical setting involves a heterogeneous patient group where patients are in need of different psychological approaches, most studies focus on specific psychological methods and specific diagnoses [14, 15]. Studies investigating the experiences of brief therapy provided in a real-world setting are thus lacking.

Studies have found that therapists are skeptical about time-limited therapy, fearing that therapy becomes superficial and less client-centered [16, 17]. Some research suggests that time-limited adaptations in therapeutic context involve greater focus on symptoms, increased

directivity, stronger containment of problems, and limitations in therapeutic approaches [17]. Therapists have reported that providing treatment within a time-limit often made them adapt their approach to the time-frame [17]. Approaches such as CBT and solution-focused approaches in time-limited treatment have been recommended due to their focus on symptom reduction [18]. Such claims suggest that treatment deadlines affect the therapeutic process in several ways, but shed little light on how this may be the case [17]. There might thus be a need for investigating how time-limitations influence the treatment processes.

Some therapists may prefer open ended treatment because of its opportunities to facilitate insight and personality reconstruction, while patients may be more content with limited treatment when they experience that it helps them feel better rapidly [11]. There still seems to be an under-researched, potential contradiction between the fixed number of therapy sessions and the possibility of involvement and activation of patients during brief therapy. The clinical practice involving customized and time-limited treatment within the context of specialized mental health services brings attention to how therapists and patients experience the new services.

Research has shown that treatment outcomes improve when patients and therapists agree on treatment goals and collaborate during treatment [19]. Since many patients have been passive recipients of treatment in the past, therapists needs to encourage them to actively participate part in their treatment by asking for feedback and to discuss treatment goals [19]. Patient activation during treatment is related to self-efficacy and self-management, as well as changes in behavior [20]. Activating patients during treatment could thus be helpful to enable them to cope with their mental health problems. Similarly, the active involving of patients in treatment is described as important to incorporate their wishes and needs, to add therapeutic value [21, 22] and to achieve efficient and effective health services [23]. The present article helps to shed light on whether patients and therapists experience the time-limit as a spur to activation and involvement, or whether it constitutes a barrier to adequate treatment. The aim of this study was therefore to explore patients' and therapists' experience of brief therapy, especially how the time frame influences the treatment process.

## Methods

This was a qualitative interview study involving patients receiving and therapists providing brief therapy in an outpatient unit in a district psychiatric centre (DPC). Inspired by phenomenology [24], we chose an exploratory approach with a broad research question that gave the informants an opportunity to shed light on the topic. This method was chosen because of the opportunity it offers in exploring both individuals' and groups' experiences. We aimed to present the results as close to the empirical data as possible, as the informants expressed themselves.

### Study setting

This study took place in a DPC in Central Norway. To provide customized treatment to young adults with anxiety and/or depression disorders, the DPC in 2016 added a specialized outpatient unit to their general outpatient services. The time-limited brief treatment program was started after a short trial-period of brief therapy provided in the general outpatient services at the DPC. The trial-period had strengthened the management's belief that brief therapy could be helpful to a larger proportion of patients if it was offered by specialized therapists working in a separate treatment unit. The DPC therefore introduced brief therapy in a new treatment unit, intending to provide early and effective help to young adults. In this study, brief therapy was defined as time-limited cognitive psychotherapy with a time restriction set at the beginning of treatment [11].

During 2019, 10 therapists provided brief therapy for 459 patients. Brief therapy was offered through up to 10 individual therapy sessions, each lasting 45 minutes. The treatment approach included CBT and metacognitive therapy [25], mindfulness [26, 27], and acceptance and commitment therapy [26]. Brief therapy was primarily aimed at patients with moderate and specific ailments, who were thought to need an effective and limited therapy approach focusing on the present. In addition to this, patients with more extensive mental health disorders could be referred to brief therapy if they could benefit from a limited focus on some of their ailments.

## Participants and recruitment

A total of 20 informants, 12 patients and 8 therapists, took part in the study, participating in individual or focus group interviews. Eligible participants were patients who were currently receiving brief therapy, and therapists who worked at the brief therapy unit. Patients were informed by the therapists about the study while receiving treatment. The therapists provided the first author's contact information for those who consented to take part. The first author then contacted these patients and made appointments for interviews. Patients were thus recruited through purposeful criterion sampling [28], i.e. those who met the study criteria. Sixteen patients consented to participate. One participant chose to leave the study before the interview took place. Patient interviews were terminated when the authors considered that the study had sufficient information power [29], after twelve interviews. The sample of twelve patients included seven men and five women, median age 31 years (range 21 to 47 years), all receiving individual brief therapy at the unit. They had been diagnosed with depression and/or anxiety by their GP prior to referral for outpatient treatment at DPC. Additionally, three patients reported also suffering from post-traumatic stress disorder. Half had been diagnosed with more than one condition, and nine out of 12 had previously received specialized psychiatric treatment. Some reported their first episode of mental ailments, whereas others had suffered from mental health problems over several decades. Thus, although all were diagnosed with depression or anxiety, the sample included patients with different backgrounds and experiences.

Therapists were recruited through purposeful criterion sampling [28], and the inclusion criterion was that they had worked in the brief therapy unit for at least six months. Eight therapists, two women and six men were eligible, and all consented to participate in focus group interviews. Their age ranged from 27 to 39 years (median 35 years), and their professional background included clinical psychologists (n = 4), psychologists (n = 3), and psychiatric nurse (n = 1). Half of the therapists had several years of experience working in the specialist health service, while the others had some experience. A total of three focus group interviews were conducted, one face-to-face and two by using Skype technology due to the worldwide pandemic outbreak in the spring of 2020.

## Data collection

Data was collected between July 2019 and May 2020. Interviews were conducted by using a semi-structured interview guide [30] where the researcher had predefined topics and then probed further as the participants responded. Representatives from the Competence center for lived experience and service development in Central Norway provided useful input to the draft interview guide, together with three of the authors (HVM, PV, MBR). The interview guide was adapted to the two participant groups, as described in the guide for interviews with patients and therapists (Fig 1). Important topics were the participants' expectations before the treatment at DPC and their experiences during treatment. The guide included questions intended to allow both patients and therapists to reflect on the

| Interview guide for patients | Interview guide for therapists |
| --- | --- |
| What did you know about brief therapy before receiving the offer? | What are your experiences with providing brief therapy for this patient group? |
| What were your expectations of the treatment in the clinic? | What are the advantages and challenges of limiting brief therapy to 10 sessions? |
| What do you think about the communication with the therapist(s)? | What does this limitation mean for whether patients take responsibility in their treatment? |
| To what extent has there been room for adapting the treatment over time? | What does this limitation mean for how and to what extent patients are actively involved in their treatment? |
| How did you experience your role in the treatment process? | To what extent can patients be involved in decision-making processes or influence the treatment they receive? |
| In what ways did you experience you were given responsibility in your treatment? | How does brief therapy compare to regular psychotherapy regarding patients' role, responsibility, and possibilities to influence the treatment? |
| What was the benefit of the treatment? | |
| What were the most important experiences with this time-limited treatment offer? | Based on your experience, what benefits do patients have from brief therapy? |

**Fig 1. Guide for interviews with patients and therapists.** A treatment strategy for meeting life as it is. Patients' and therapists' experiences of brief therapy in a district psychiatric centre—a qualitative study.

treatment process including their responsibility throughout the treatment process, and thoughts concerning time-limits of the treatment. They were asked about their experiences with communication between therapists and patients and what role both groups had played in the treatment process. Moreover, they were asked about their perspectives on the advantages and disadvantages of time limited treatment.

The first author (HVM) conducted all the individual patient interviews. All interviews were conducted at the university, according to the patients' wishes. Interviews with patients lasted from 40 to 60 minutes. The first (HVM) and last author (MBR) conducted the focus group interviews with therapists at the clinic together. The Skype-interviews were led by the first author (HVM) from her home-office, and the participants attended from their home offices due to Covid-19 government restrictions. Due to illness, one of the therapists was interviewed individually over Skype a few days later. Interviews with therapists lasted from 40 to 80 minutes.

The interview-guide remained unchanged, with the exception of questions to explore whether patients were familiar with this type of treatment beforehand; whether it was possible to adjust the treatment program during treatment; and whether any part of the therapy had been different compared to what they had expected. All interviews were audio recorded and transcribed verbatim. The accuracy of the transcriptions was checked by the first author by listening to all audio recordings several times. Notes on first impressions after each interview were also logged by the interviewer(s) and contributed to the analytic process.

## Data analysis

Data analysis was conducted using the method of Systematic Text Condensation (STC) [24]. STC is a phenomenological methodological approach that aims to describe the informants' experiences, as expressed by themselves. The analysis was conducted in a group of five researchers with different backgrounds (medicine, psychology, social sciences, health services research). HVM is a doctoral candidate in medicine with a professional background from medical sociology and community planning at MA-level and several years of experience from mental health hospital planning including service innovation tasks from this area of Norway. LA is a researcher and physician (specialist in physical medicine and rehabilitation). PV is an associate professor in health sciences and health services researcher. BH is a researcher and associate professor in Health and Care Science. MBR is a researcher and professor in mental health work with a background from psychology and public health.

The data analysis was conducted according to the four stages of STC [24]. All interviews were initially read by the first author (HMV), all therapist interviews were also read by MBR, whereas patient interviews were divided between MBR and PV. First impressions resulted in preliminary themes which were discussed. Examples of preliminary themes at the first step were "tools to deal with mental illness and not focus on emotions", "influence on own treatment", and "the responsibility lies with me".

Secondly, unit of meanings [24] were identified and coded. The coding was conducted by systematically examining the text line by line, looking for meaning units that could shed light on the aim of the study. All codes were collected in a coding list by HVM, according to empirical coding [31], as close to the empirical material as possible [24]. The coding list was discussed by the authors, sorted in code-groups and grouped into categories, representing the main themes in the material. The code groups appeared, for example, as "experience of the treatment offer", "experience of participation and activation" and "advantage of the treatment versus need for more time". The focus group interviews with the therapists were coded [31] by HVM. The codes from these interviews were grouped into categories, representing the main themes in the interviews with the therapists. Examples of categories from therapists were "patients' benefits", "termination competence and limited focus" and "activation".

Thirdly, the main categories from the interviews with the patients and therapists, respectively, were collected, compared and collated, before condensing and summarizing the main themes in the overall data material. Based on all categories, themes were constructed and validated through discussions in the complete author group. Empirical dimensions were formed for each interview and similarities across interviews were reflected in the themes. In the fourth step, descriptions and concepts reflecting the informants' most important experiences were synthesized. The process continued until data reached a point of convergence, resulting in five themes that encompassed the data material. Before agreeing, all the authors looked for alternative interpretations in discussions.

Preliminary findings were also presented and discussed in meetings in a research group on patient education and participation where HVM and MBR are members. Quotes from the

data material were chosen to elaborate and illustrate the results. They were translated by LA and checked and approved by MBR and HVM.

## Ethics

This study was approved by the Regional Committee for Medical and Health Research Ethics (REK) in Central Norway (2018/49). REK are composed of persons with different professional backgrounds, lay representatives and representatives of patient associations. The seven regional committees are appointed by the Ministry of Education and Research in Norway for four years at a time. Patients received oral and written information about the study at the clinic and their signed consent was obtained. The first author received the therapists' contact information from the clinic manager before interviews took place. Therapists received oral and written information about the study at the clinic and signed a written consent. This project was conducted in line with the Helsinki declaration (World Health Organization, 2010).

## Results

A total of 20 informants, 12 patients and eight therapists, took part in individual or focus group interviews. The patients were currently receiving brief therapy, and the therapists worked at the brief therapy unit.

The main results constitute five themes; 1) Time-limit as a frame for targeted change, 2) Clarifying expectations and accountability, 3) Shared agreement on a defined treatment-project, 4) Providing tools instead of searching for causes, and 5) Learning to cope—not being cured.

The main findings of the five themes are summarized in the figure Themes with key aspects (Fig 2).

### Time-limit as a frame for targeted change

Time-limitation was described by the therapists as a cornerstone in brief therapy. According to them, the time frame helped them to stay focused on the patients' change and maintain structure and focus during treatment. The patients' acceptance of the time-limit was described by the therapists as crucial to benefit from treatment. When starting treatment, they informed the patients that they had to accept and trust that sufficient treatment was possible within this time frame. Although most patients said they had not been aware of the time-limits before starting treatment, they accepted it and did not see it as problematic. They perceived the time-limit for the treatment as adequate and said it helped them look differently at their mental problems, describing it as a signal to them that their mental illness did not have to last a lifetime.

*[. . .] it gives the picture that mental illness doesn't have to be an eternity project. That it is possible to succeed [. . .]The time-limitation contributes to thinking of [my problems] as a normal illness.* (Female patient no. 6)

Several therapists experienced that the predefined time-limit for treatment made it easier for them to focus on providing structured treatment. The time frame forced them to evaluate each step during treatment. When comparing brief therapy to regular outpatient treatment, the therapists described the brief therapy process as more focused and thereby more predictable. The time- limitation enforced clearly defined treatment goals.

*[The time-limitation] makes it easier for me to be structured. It is important for me not to waste time. . . I would rather not waste a minute of any session. Fifteen minutes passing without me knowing how I used them [. . .] for the direction I am going in, that is not good. So,*

| Themes | Key aspects[1] |
|---|---|
| **Time-limit as a frame for targeted change** | Requirements: acceptance of time-limitations; planning and closure skills; early termination if found unwanted/ineffective<br>Consequences: clearer goals; increased structure and focus; stepwise assessment; increased predictability; signaling optimism concerning the duration of problems; strengthening optimism<br>Challenges: insufficient time for both assessment and treatment |
| **Clarifying expectations and accountability** | Requirements: joint clarification of expectations; patients' assuming responsibility for and being actively involved in treatment, including between therapy sessions<br>Consequences: increased self-efficacy after treatment; faster change<br>Challenges: lack of motivation; concerns about relapse; more work after completion of therapy |
| **Shared agreement on a defined treatment-project** | Requirements: agreement on time-limitation; including a clear treatment plan aiming at concrete, realistic and limited goals; agreement on how to work as a therapist-patient team<br>Consequences: more structured therapy; experiencing therapy as meaningful and time-efficient; aiming for patients to take ownership of their process; hope for improvement |
| **Providing tools instead of searching for causes** | Requirements: patients' awareness of their problems; active engagement in own therapy; using tools in between therapy sessions<br>Consequences: increased active participation; learning to use self-management tools; dealing with negative thoughts and stress; improved mastering of life |
| **Learning to cope – not being cured** | Requirements: aiming to "master life as it is" and to support self-management; changing focus towards modifying previously unfortunate strategies<br>Consequences: strengthened self-efficacy and coping ability<br>Challenges: continuing the change-work on their own |

**Fig 2. Themes with key aspects.** 1. The different key aspects describing themes could include requirements, consequences, and challenges.

*you need to focus all the time. Know that what you are doing is useful for where you are, and where you are going. We must evaluate continuously, to know that we are on the right track.* (Therapist no. 1, group no. 3)

The therapists described that the time-limitation entailed focusing on closure when starting treatment, and that closure was a theme discussed with patients throughout the sessions. Several therapists compared brief therapy to ordinary outpatient treatment and pointed out that brief therapy to a greater extent required good "closure skills". Having termination in sight during treatment was described as crucial to stay focused.

*[. . .] closure competence is about what I say when the patient arrives. It makes it much easier for me to repeat if I realize that we are at that point [treatment approaching conclusion]. In addition, it makes it much easier for the patient to accept when I say: Do you remember that we spoke about this during the first session?* (Therapist no. 1, group no. 3)

Several patients found that therapists were good at explaining the time-limited treatment process and therapists' expectations of them when starting treatment. They said they understood early on that the treatment might be challenging but efficient, and they felt motivated to undergo the treatment.

*Yes, [time-limited] treatment is challenging, but at the same time I feel. . . there is no coercion, I don't feel like having to do anything. I do this because I want to achieve something. It is challenging, but that is what I need.* (Male patient no. 9)

If the therapists later in the process saw that treatment was ineffective, they could also terminate treatment early (after 3–4 sessions). They said that it was important to provide this treatment for those who wanted it or could benefit from it, instead of those who were not receptive or did not accept the time frame. Some therapists found it was challenging to both conduct an assessment and provide adequate treatment within the limit of 10 therapy sessions. They were concerned that the quality of either could be compromised and suggested separating assessment and treatment more clearly from each other. Patients, on the other hand, emphasized the trust that was established between therapist and patient during the assessment and that this laid the foundation for the targeted treatment.

According to the therapists, time-limitation was clearly defined, and any extension had to be discussed and agreed within the therapist team. In contrast to this, most patients said that, although brief therapy was limited, they were confident that they would receive more treatment if they needed it.

*But. . . if it becomes clear that I need more sessions, for some reason, I expect that I will receive that [. . .] You can`t stop building when the house is built halfway, can you*? (Male patient no. 12)

## Clarifying expectations and accountability

Both therapists and patients described the treatment as based on a joint clarification of expectations, committing both parties during therapy and making the patients accountable for their part in the treatment process. The therapists described it as establishing "a contract" with the patients; a concrete mandate for treatment, enabling patients for the treatment process. Patients' commitment was described as crucial for progress during treatment, and progress depended on patients believing that the treatment would work for them. If patients did not "feel ready" for this treatment, they were advised to seek help elsewhere, or wait until they felt ready to work actively on their problems.

*It is about making patients accountable through clarifying expectations [. . .] You cannot go to the psychologist to vent and then improve. It requires hard work and spending time sensibly. The goal is to generate change.* (Therapist no. 3, group no. 1)

The patients said they felt responsible for doing most of the work in therapy, and that they were expected to play an active role in their treatment. They had to show willingness to work on their problems in between the weekly sessions, and that making progress was their responsibility. Patients and therapists agreed that for therapy to work, patients had to be willing to work hard on their problems. Some described this as the "ball being in their courtyard". They could not be passive recipients of the treatment, as they could not assume the therapist could "fix" them. Instead, patients had to adjust their expectations to what the brief therapy could offer and have the determination and willingness to work hard to change their previous strategies.

*In the beginning, I had the same expectations as when going to the doctor, that someone else can fix the problem [. . .] Gradually, I realized that it is not quite like that. But that was my*

*first expectation–to go to the doctor to be repaired. My previous experience with the health services is that if you have [problems], you go there, and it is fixed.* (Male patient no. 2)

The therapists explained that their treatment was mainly inspired by cognitive therapy, and that it was not uniform, or manual based. They reported that patients had to have a specific attitude when receiving short treatment, since the treatment was "no place for rest" but rather required activation. They thought the treatment should be demanding and that change would not be possible if patients were not accountable and highly motivated.

*If you are going to jump over an abyss or something similar, it is not smart trying to do this in three small jumps. You must make the big jump. If you try to do this gradually you will have a problem. So. . . this could be a way of selling it to the patients [. . .] the most important is that they themselves must want to make the jump.* (Therapist no. 2, group no. 3)

Several patients confirmed the notion that brief therapy required a changed mindset. They took on greater responsibility for doing something about their problems by looking at it from a new perspective. Some patients reported that the treatment had changed the way they dealt with their problems and they believed that their own efforts would help change their everyday lives after the end of treatment. Others said they had changed more than expected, in a very short time. However, several patients experienced this development as a fragile beginning of changing strategies in life, and that they were afraid of relapse. They underlined that personal change would require more time and work after completing therapy.

*Of course, the responsibility is mine. . . but I feel that it is in a way based on my capabilities. I become responsible for trying to challenge myself. Every week [. . .] I receive homework that I am expected to do. And then I am responsible for doing them every week.*

(Male patient no. 11)

## Shared agreement on a defined treatment-project

Both patients and therapists described brief therapy as focusing on a time-limited project, i.e. collaborative treatment that was carefully planned to achieve an aim within a specific period. The patients were not supposed to change their entire lives. Patients described the treatment as a structured, "forward-looking" and intensive training program, and as a "change project" that helped them target and refine their own efforts towards the endpoint of the treatment. They experienced brief therapy as meaningful, because they did not waste any time on something that did not work, like extensive talks about their feelings or their existential pain.

*We define a project. . . decide what exactly we are working on. [. . .] To make it focused. And he [the therapist] brings it back whenever it sideslips and drags me back. It makes me aware of exactly where I get lost or what I blame. . . this is the focus now. For better or worse. And I kind of like it. Because it gives me the feeling of having a concrete goal to work towards.*

(Female patient no. 5)

The therapists described brief therapy as defined by a beginning and a predetermined end, with a clear treatment plan aiming to achieve concrete goals. According to them, a clearly defined project helped making the treatment more structured. They considered it a concrete and targeted treatment, where patients' ailments might be normalized. They linked this to the

patients' expectations, and the project described what the patient was supposed to work on and what the results should be.

> *Since we have limited time, we have to define a project clearly [. . .] the patients need to know why they are here, what they are working with, and what expectations they can have [. . .] what effect it should have [. . .] It makes you more focused. We are conscious of the number of sessions available, and we must make a plan together with the patients.* (Therapist no. 3, group no. 2)

According to the therapists, they aimed for patients to take ownership of their therapeutic project, based on realistic expectations of what they could achieve within a short period. Therapists and patients had specific roles during the treatment process and developed a shared understanding of how to work as a team to achieve the patients' goals. Patients said that they felt confident in the therapists' role and competence, contributing to their sense of hope for improvement.

## Providing tools instead of searching for causes

Both patients and therapists described that the brief therapy process did not entail investigating the causes of patients' problems. Instead, several techniques and "tools" were introduced and discussed. The therapists explained that many patients had searched for the causes of their problems in the past, to find answers as to why they were depressed or anxious. According to therapists, patients' perspectives of their problems were crucial; and taking a more active role and using appropriate techniques would be more helpful than searching for causes. Tools and exercises could for example include "awareness training", allowing patients to see that even when life did not seem so good, they could change the way they viewed their problems. Patients' active use of tools in between treatment sessions was considered essential for learning.

> *Often, I use the physiotherapist comparison. . . that we can demonstrate good exercises, but this doesn't help if you don't practice the exercises between our sessions. Practice is the only thing that gives results. There is no use going see a physiotherapist for one hour a week to improve. You must practice in between.* (Therapist no 2, group no. 3)

Many patients referred to their therapists as teachers, coaches, personal trainers or mentors. They did not feel like traditional patients in treatment, but rather as active students learning how to use tools. Some said that the brief therapy unit was like a fitness center, where you had to work hard and do the exercises regularly to get results. The patients said they had learned techniques for dealing with negative thoughts and stress, and they had found that simple tools had contributed to a greater change in mastering life than they had expected.

> *I am more conscious about things. And I have received tools. I do not have the whole toolbox yet, but. . . maybe, earlier I received the whole toolbox but without tools in it. Now I have a spanner and a hammer. . . and I might need a saw or something like that. And I might eventually start to think that I might not need a power tool. That I can use the screwdriver I received during treatment.* (Female patient no. 3)

### Learning to cope—not being cured

According to both therapists and patients, the search for causes entailed a wish to find a definitive cure for the mental health problems. The therapists described brief therapy as having a different aim, namely, to learn ways to "master life as it is". Instead of searching for causes and a cure, they wanted patients to change their focus towards modifying previous strategies. Several patients described previous experiences with mental health treatment and said they were pleased that brief therapy used a different approach. Some pointed out that the search for causes of their ailments could take a long time and they found brief therapy more helpful in finding concrete solutions to strengthen their coping skills.

*During some [more extensive] previous treatments I quit. I couldn't stand all the deep dives into all that pain.* (Female patient no. 5)

*This makes more sense. I don`t feel that I waste time going there, because I bring something home. Instead of just talking about how I am feeling. I can do that with my friends.*

(Female patient no. 3)

The therapists emphasized that brief therapy did not entail supportive talking about everything the patient wanted or needed to discuss. Instead, they focused on what could lead to a more constructive understanding of the patients' ailments and that patients could be able to cope with their problems without necessarily becoming symptom-free. The main aim was to enable patients to continue their recovery process on their own; to be "their own therapist".

*People need to understand that it is not me helping you. We discuss what they need to change, how they should relate differently to things. And then they need to implement it themselves. We try to build them up, also their confidence. . . They have the possibility to make changes in their lives. Most patients come with a feeling that their freedom and willpower are reduced by symptoms. When they change that. . . they are the ones getting out of it. Then you can become your own therapist.* (Therapist no. 3, group no. 1)

Therapists said that they aimed for a "change-process" that would continue after the treatment was completed. They wanted patients to understand that they could choose to change the life course and wanted to enable them to continue the process themselves. The therapists wanted patients to understand that challenges did not need to be part of their identity in the future, and that it was possible to manage life better. According to the therapists, patients were not expected to become symptom free during treatment. The goal was to change the patients' understanding of the challenges and provide them with tools enabling them to continue the change on their own. One of the patients illustrated how his perspective towards mental health problems had changed:

*Here, I feel like I am the boss. My mental health problems should not manage me, I should manage my life. Through them [my problems] or to live with them. Without anything ruining my everyday life.* (Male patient no. 8)

## Discussion

The results showed that time-limits in therapy was experienced as a cornerstone in brief therapy. Time-limits were perceived as an explicit framework for targeted change, which helped

both therapists and patients maintain structure and focus during the treatment-process. A joint clarification of expectations, committed both parties during therapy and made the patients accountable for their part in the treatment process. Brief therapy was described as a collaborative treatment that was carefully planned to achieve individual aims in a defined treatment project. The purpose of the treatment was explained as mastering ailments, not getting rid of them. Both patients and therapists explained that it was more about dealing with ailments using tools and techniques and less about finding the causes of the ailments. The therapists explained that the basic idea was that patients could be better able to cope with life as it is, and then to have a greater opportunity to manage life on their own after the end of treatment.

## The time-limited therapy process

In the present study both patients and therapists described the therapy as defined by a time-limited framework that helped them shape a shared treatment plan. While an understanding of the effect of time-limits on the therapeutic process is important, it is difficult to examine due to several interacting variables characterizing treatment practice [17]. While it has been suggested that many therapists consider that time-limit does not affect therapy [11], the present study described the time-limit as a cornerstone in brief therapy, helping keep the focus on the patient's personal goals during treatment. Time constraints have been shown to affect both the outcome of treatment and the therapeutic process [11, 17], exerting pressure on the therapy process which creates an expectancy effect for both patients and therapists, with both positive and negative consequences [11]. In the present study, the time-limit forced the therapists to evaluate each step during treatment, making the treatment focused and predictable. An imposed time-limit also has the potential to threaten the therapeutic alliance [17], described as empathy, warmth and acceptance in therapy [32]. Therapeutic alliance is viewed as indispensable in all forms of therapy [33] and as crucial for successful goal setting and treatment [33], and is a robust predictor of psychotherapy outcomes even in brief therapy such as guided internet interventions [34]. In the present study the patients emphasized that trust between therapist and patient was established early in the treatment process, potentially having a positive influence on the treatment process.

In our study, the therapists underlined that brief therapy strengthened the therapists' termination competence and played a role in creating a more positive attitude to treatment closure for the patients. In contrast, research has shown that therapists feel that their way of working is altered by time-limits [11] and termination of treatment in psychotherapy has been recognized as emotionally difficult for both clients and therapists [35]. Patients' reactions to termination are related to their evaluation of the treatment process in the span between experience of success and regression, and can be expressed as a need for further treatment [36]. According to our study, the time-limitation entailed focusing on closure when treatment started and made the treatment process more predictable and focused. Studies have showed that therapists react differently to termination of treatment, dependent on whether the termination is planned or not [35, 36]. Therapists react with more pride when a termination is planned, and with anger, mourning, frustration and anxiety when unplanned termination occurs [35]. Positive emotions are especially related to the therapists' experience of achievement and pride in the clients' success when termination was planned [35].This is in line with our findings, where therapists said it was easier to end treatment when the time for termination was set in advance. Our findings indicate that the time-limited framework might play a positive role in therapists' sense of control over the therapeutic process.

In the present study, patients expressed that they had to accept the time-limits set by the therapists, and that they had to commit and be ready for change. This is in line with research

showing that time-limits can be associated with therapists taking on a more directive or paternalistic role [11]. In our study, this was expressed through the signal to patients that they would not get any treatment if they signaled unwillingness or inability to participate in the limited-time treatment. This may indicate a signal that patients should believe in time-limited treatment to achieve therapeutic results. On the other side, time-limits and continuity in therapy can be experienced as positive and predictable for patients in a period where they must undergo significant changes in life [37]. Some of the patients in the study emphasized that the time constraint felt challenging, but that it had helped strengthening the focus on changing strategies during treatment and was something they needed. A therapist may be described in different ways, partly as a competent expert and partly as a safe point that provides emotional support during a turbulent change process [37]. In our study, patients expressed confidence in the therapists' competence and trusted they could help them acquire coping-strategies within the fixed time frame. They elaborated that the predictable time frame had helped to improve their ability to cope with ailments more than they had expected before treatment. The findings may indicate that a planned and fixed time-limit in brief therapy helped to strengthen patients' motivation during treatment.

Contrary to this, the results also showed that most of the patients believed they would receive more treatment if necessary, after ending brief therapy. This means that therapists and patients perceived the fixed time frame differently. This should be viewed in the context of the comprehensiveness of the Norwegian welfare system. In this context, it can be assumed that patients expect to receive help if they need further treatment.

## Patients' readiness and activation

Even though the patients were entitled to general outpatient treatment at the hospital level (DPC), our results showed that both therapists and patients experienced that patients had to have certain resources to be able to take an active patient role in brief therapy. People seeking psychotherapy come to treatment with different motivation, preparation and capacity for behavior change [38]. Patients' readiness to change behavior greatly affects the process and outcome of treatment, and patients can be at different stages of such readiness before therapy takes place [38]. Given a variability in patient readiness to change, uniform time-limits for treatment would not adequately serve all the patients' needs, even though their diagnoses are similar [7]. Patients' expectations before treatment might predict the outcome, and these should therefore be identified to ensure that the treatment satisfies the patients' needs [39]. In our study, few of the patients knew in advance what the treatment entailed. The therapists assessed patients' readiness before starting treatment, emphasizing that they only offered treatment to patients they believed had the ability to take responsibility for their own treatment process. This finding indicates that assessment of the patients' readiness before treatment can be viewed as mapping the potential to patient activation during brief therapy.

The fact that the patient should play an active role during treatment, has become a central goal in mental health treatment [40]. Various terms have been used to describe aspects of this goal, such as patient participation and shared decision making [40]. Patient participation is claimed to be an important element in high-quality mental health services [41] and research has emphasized the need to increase the focus on patients' possibility of influencing treatment by their preferences and values [41]. Better results can be expected when patients and therapists agree on therapeutic goals and the processes for achieving those goals [19]. Recent research has highlighted the need for continuous learning, patient co-production and more efficient care in mental health care, as well as emphasizing the need for mutual learning between therapist and patient during the treatment process [42]. In the present study we

found this described as a clear and reconciled treatment plan aiming to achieve concrete and individual treatment goals through the patients' active participation. The patients expressed that the treatment was based on a shared understanding of expectations, even though the patients could not influence the general time frame or method of treatment. Results of psychotherapy seem to improve significantly when the patient and therapist are actively involved in a collaborative relationship [19]. In the present study, the experiences of shared understanding may thus have reinforced patients' positive expectation of achieving their individual goals through brief therapy.

Also previous studies on brief therapy among young adults have described patients perceiving their therapists as experts or teachers who could give them a new understanding of themselves and their thought patterns, and help ailments appear more understandable and manageable [37]. The patients may see their therapist as an expert with an objective perspective who could shed light on their difficulties [37] and the therapy can be about facilitating self-acceptance of one's own illness or ailments and for dealing with associated consequences [39]. Brief therapy in the present study was strongly influenced by CBT, where both the therapist and the client are seen as experts in their own specialist areas, and work together to create change [43]. The structure and process associated with cognitive therapy has also been shown to contribute to increased adherence to homework given during treatment, and patients have gradually learned to become their own therapists [43]. This is in line with the present study, where patients actively used "treatment tools" between treatment sessions and where responsibility for doing this was emphasized as crucial for learning and further personal effort. Patients also expressed feeling challenged by the treatment process because they knew the results depended on their own efforts. Both patients and therapists emphasized the patients' responsibility in their own treatment. This finding highlights the importance of patients' ability to activate themselves and may therefore be viewed as an important part of brief therapy. It can be assumed that brief treatment preferably should be offered to patients who are prepared and express that they feel ready for activation.

## Mastering life as it is

In our study, both patients and therapists expressed that a main purpose of treatment was that patients should be able to master life as it was. The patients emphasized that the therapy made them understand that the ailments did not need to develop into a chronic or long-term condition, and they had experienced that therapeutic tools could be learned and used to better cope with their ailments. Facilitating personal development during treatment can be linked to a recovery-oriented approach [40]. The recovery perspective in mental health treatment has two core principles, that people with mental illnesses can lead productive lives even while having symptoms, and that many will recover from their illnesses [44]. This approach recognizes that people with mental ailments or illnesses might be able to return to and participate in society [44]. The patients in this study described a positive change in their perspectives, seeing the treatment as the beginning of a longer recovery process they had to continue by themselves. This is in line with the recovery-oriented focus on restoring individuals' functioning beyond symptom reduction [44], such as developing skills and knowledge that they need to take personal responsibility for their health [45]. This is also in line with CBT which focuses on the future, as opposed to the past, and on the participants' strengths and abilities, as opposed to their problems and shortcomings [46]. Personal recovery in mental health has been defined as a personal, unique process of changing one's attitudes, values, feelings, goals, skills and/or roles [47]. Therefore, the patients' expression of brief therapy as a fragile beginning of a longer improvement process might be viewed in the perspective of personal recovery.

The patients in our study described brief therapy as a "change project" making them look at their problems in a new perspective. Similarly, the therapists expressed focusing on ways patients can live a satisfying, hopeful and contributing life. Hope is described as one of the most important factors concerning recovery, and whether one believes that recovery is feasible [48]. Patients' hope is also an important factor in therapy that therapists can help promote and strengthen [32]. Our findings indicate that the patients' hope for recovery after the treatment process were reinforced by clinicians expressing belief that brief treatment could be tailored to them and their individual needs. This is in line with research claiming that hope can enhance motivation to engage in the recovery process [48]. Expectations to benefit from treatment can be regarded as the activating energy of hope, and the therapist also might increase a patient's sense of expectation [32]. This hopeful expectation can again be nurtured by the therapists [32]. Experience of hope might strengthen the patients' focus on changed thinking and behavior in the recovery processes, including strengthened optimism about the future [47]. These findings indicate that brief therapy aimed to be both therapeutic and prevent future problems. The ending of brief therapy might therefore be viewed as a starting point of a personal recovery-process aiming to master life as it is.

## Strengths and limitations

The sample consisted of both patients and therapists and variation in age and gender, constituting a strength of this study. In addition, the therapist sample included most of the therapists working in the treatment unit, having various length of experience with brief therapy. A potential limitation is the apparent homogenous therapist group, due to the management recruitment of professionals to the brief therapy unit. Here, those who were especially interested in brief therapy were encouraged to apply, thus being dedicated to brief therapy. The patient sample included a breadth of patient experiences, both patients who experienced mental health problems for the first time and some with recurring episodes over decades. This enriched the data material. A potential limitation is the possibility of a sample bias through voluntary recruitment. The patients were recruited via the therapists, and all the therapists in the unit was supposed to recruit patients who had some experience of receiving brief therapy. The therapists' presentation of the study might have influenced who agreed to participate. Nevertheless, the study depended on the study participants who had first-hand experience with receiving brief therapy. We therefore found this part of recruitment process adequate.

The qualitative semi-structured interview approach strengthened the exploratory approach. The interviews with patients were conducted when they were currently receiving brief therapy, or shortly after the end of treatment. Interviews at a single point include a risk of bias, since the interviews were conducted before patients knew the outcome of their treatment. A longitudinal approach should be pursued in further research on the possible personal recovery process that may take place after brief treatment ending.

As the analysis was conducted by five authors with different professional backgrounds, the researchers had complementary experiences which strengthened the analytical processes. Discussions in a different research group also strengthened the study through providing alternative considerations.

A limitation might also be that both therapists' and patients' experiences and point of view are not confirmed by other types of data in this study. In addition, the Norwegian socio-cultural context, such as the Norwegian health care system and the comprehensiveness of the welfare system, somewhat limits the transferability to other countries. That patients expected to receive more help from the welfare system if they needed further treatment might rest on the Norwegian welfare system.

## Conclusion

This study adds knowledge on patients' and therapists' experiences with time limitations during therapeutic processes. The time-limitation appeared to play a positive role, helping the therapists to structure the therapeutic process and strengthening the patients' motivation. An experience of a shared treatment project and the opportunity for active participation in treatment may have reinforced patients' responsibility and positive expectations to achieve individual goals. This emphasizes the importance of the patient's readiness and activation as important prerequisites for brief therapy. In addition, brief therapy was perceived as the beginning of a longer recovery process patients had to continue by themselves. It can be assumed that the therapists'efforts to nurture the patients' hope for improvement during treatment contributed to increasing the motivation for the further efforts and recovery process after the end of treatment. The end of brief therapy can be viewed as a fragile and uncertain start to the patients' process towards "mastering life as it is". More research is needed to investigate the patients' long-term outcome after such treatment and to shed light on the potential for, and limitations of, mastering everyday-life after ending brief therapy.

## Supporting information

**S1 Fig. Poster presentation; A TREATMENT STRATEGY FOR MEETING LIFE AS IT IS, the results from the present study, presented as preliminary findings to colleagues at «ECMH–European Conference on Mental Health 2020» Online conference, Finland, October 2020.**
(TIF)

## Acknowledgments

We thank all the therapist's and patients' for taking part in the interviews. We also thank the representatives from the Competence Center for Lived Experience and Service Development in Central Norway for useful input to the research project.

## Author Contributions

**Conceptualization:** Hilde V. Markussen, Marit B. Rise.

**Data curation:** Hilde V. Markussen, Marit B. Rise.

**Formal analysis:** Hilde V. Markussen, Marit B. Rise.

**Funding acquisition:** Hilde V. Markussen, Marit B. Rise.

**Investigation:** Hilde V. Markussen, Marit B. Rise.

**Methodology:** Hilde V. Markussen, Marit B. Rise.

**Project administration:** Hilde V. Markussen.

**Resources:** Hilde V. Markussen.

**Software:** Hilde V. Markussen.

**Supervision:** Lene Aasdahl, Marit B. Rise.

**Validation:** Hilde V. Markussen, Lene Aasdahl, Marit B. Rise.

**Visualization:** Hilde V. Markussen, Petter Viksveen.

**Writing – original draft:** Hilde V. Markussen.

**Writing – review & editing:** Hilde V. Markussen, Lene Aasdahl, Petter Viksveen, Berith Hedberg, Marit B. Rise.

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
