## [Decision Letter · Decision Letter 0]

24 Sep 2021

PONE-D-21-16580A treatment strategy for meeting life as it is. Patients’ and therapists’ experiences of brief therapy in a district psychiatric centre - a qualitative studyPLOS ONE

Dear Dr. Markussen,

Thank you for submitting your manuscript to PLOS ONE. After careful consideration, we feel that it has merit but does not fully meet PLOS ONE’s publication criteria as it currently stands. Therefore, we invite you to submit a revised version of the manuscript that addresses the points raised during the review process.

We look forward to receiving your revised manuscript.

Kind regards,

Frédéric Denis, Ph.D.

Academic Editor

PLOS ONE

2. Thank you for stating the following in the Acknowledgments/Funding Section of your manuscript:

“The Liaison Committee between the Central Norway Regional Health Authority and Norwegian University of Science and Technology (NTNU) funded the work under Grant no. 22314. The funding body was not involved in the design of the study, the collection, analysis and interpretation of data or in writing the manuscript.”

“The first author (HVM) and the last author (MBR) received the funding through their affiliation with NTNU and St.Olavs Hospital.

The Liaison Committee (No; Samarbeidsorganet) between the Central Norway Regional Health Authority and Norwegian University of Science and Technology (NTNU) funded the work under Grant no. 22314.

The funder website: https://helse-midt.no/samarbeidsorganet

Additional Editor Comments (if provided):

This manuscript aims to explore patients' and therapists' experiences of brief therapy. It is a timely and well-written article that nevertheless raises some minor comments.

1/P 170 : « The patients reported suffering from depression and/or anxiety, and/or post-traumatic stress disorder ». A medical evalation has it been carried out beforehand ?

2/ Could you present the interview grid

3/ Please give more information about your participants (age, gender...).

Reviewers' comments:

Reviewer's Responses to Questions

**Comments to the Author**

1. Is the manuscript technically sound, and do the data support the conclusions?

Reviewer #1: Yes

2. Has the statistical analysis been performed appropriately and rigorously? 

Reviewer #1: Yes

3. Have the authors made all data underlying the findings in their manuscript fully available?

Reviewer #1: No

4. Is the manuscript presented in an intelligible fashion and written in standard English?

Reviewer #1: Yes

5. Review Comments to the Author

Reviewer #1: This is very well written manuscript that explored patients' and therapists' experience of brief therapy, especially how the time frame influences the treatment process, using a qualitative methodology.

I think that the paper brings interesting evidence in addition to the literature.

It would be clearly improved by adding a Table that summarizes the main findings of the five themes in a structured manner highlighting the main ideas.

6. PLOS authors have the option to publish the peer review history of their article (what does this mean?). If published, this will include your full peer review and any attached files.

Reviewer #1: **Yes: **Wissam El-Hage

---

## [Author Response · Author response to Decision Letter 0]

8 Oct 2021

Please read the "response to Reviewers".

We have answered all questions.

---

## [Editor Report · Decision Letter 1]

11 Oct 2021

A treatment strategy for meeting life as it is. Patients’ and therapists’ experiences of brief therapy in a district psychiatric centre - a qualitative study

PONE-D-21-16580R1

Dear Dr. Markussen,

We’re pleased to inform you that your manuscript has been judged scientifically suitable for publication and will be formally accepted for publication once it meets all outstanding technical requirements.

Kind regards,

Frédéric Denis, Ph.D.

Academic Editor

PLOS ONE
---

## [Editor Report · Acceptance letter]

15 Oct 2021

PONE-D-21-16580R1 

A treatment strategy for meeting life as it is. Patients’ and therapists’ experiences of brief therapy in a district psychiatric centre - a qualitative study 

Dear Dr. Markussen:

I'm pleased to inform you that your manuscript has been deemed suitable for publication in PLOS ONE. Congratulations! Your manuscript is now with our production department. 

Kind regards, 

on behalf of

Dr. Frédéric Denis 

Academic Editor

PLOS ONE